# Molecular Real-Time PCR Monitoring of Onion *Fusarium* Basal Rot Chemical Control

**DOI:** 10.3390/jof9080809

**Published:** 2023-07-30

**Authors:** Elhanan Dimant, Ofir Degani

**Affiliations:** 1Plant Sciences Department, MIGAL—Galilee Research Institute, Tarshish 2, Kiryat Shmona 1101600, Israel; elhanand@migal.org.il; 2Faculty of Sciences, Tel-Hai College, Tel-Hai, Upper Galilee 1220800, Israel

**Keywords:** *Allium cepa*, crop protection, fungus, host-pathogen interactions, *Fusarium acutatum*, *Fusarium oxysporum*, pathobiome, pesticide, Prochloraz

## Abstract

*Fusarium* basal rot disease (FBR) is a destructive threat to onion crops around the globe. It causes seedlings’ death, development disruption, and pre- and post-harvest bulb infection and rotting, with a concern for toxin infestation. It is an emerging disease in Israel, with new reports from farms nationwide. Recently, we reported on a full-season pot experiment to protect two leading commercial cultivars against FBR chemically. Here, we present new real-time qPCR molecular tracking of the pathogens inside the host plant and compare the infection levels to a deep analysis of the impacts of this experiment’s treatments on plant growth and health indexes. The new findings reveal variations within each treatment’s effectiveness regarding sprout development and bulb ripening stages. For instance, in the yellow Orlando cv., high protection was obtained with Azoxystrobin + Tebuconazole (Az-Te) in sprouts against *F. oxysporum* f. sp. *cepae* and with Fludioxonil + Sedaxen in mature plants against *Fusarium acutatum*. Thus, combining these fungicides may protect plants throughout their lifecycle. Also, Prochloraz at low dose was highly efficient in the Orlando cv. Still, to shield red Noam cv. plants from both pathogens, increasing this fungicide concentration towards the season-ending should be preferred. The qPCR tracking showed that all chemical treatments tested could reduce infection from pathogens by 80–90%, even with compounds such as Az-Te that were less effective. This implies that the pesticide was effective but probably phytotoxic to the plants, and thus, lower dosages must be considered. The molecular-based analysis discloses the high infection ability of *F. oxysporum* f. sp. *cepae* compared to *F. acutatum* in both cultivars. It also indicates an antagonism between those species in the Orlando cv. and synergism in the Noam cv. The current work reveals weak and strong points in chemical FBR protection and offers new ways to improve its application. The qPCR-based method enables us to closely monitor the pathogenesis and efficacy of chemical-preventing treatments and optimize crop-protection protocols.

## 1. Introduction

*Fusarium* basal rot disease (FBR) causes substantial yield losses in onion (*Allium cepa*) crops across various regions globally [1]. Despite the widespread occurrence of FBR in different species of *Allium*, our understanding of this disease remains fragmented and incomplete [2]. The most frequently reported species responsible for this disease is *Fusarium oxysporum* f. sp. *cepae*, which explicitly affects onion cultivars. Nonetheless, the condition can also be caused by other *Fusarium* species [2].

An onion plant fungal infection typically occurs through the roots and causes rotting of the basal plate (where the roots and stem are connected). The disease can lead to seedling mortality before or after aboveground emergence, known as “damping off”. Infected sprouts may exhibit disrupted growth, yellowing leaves, and total wilting. Yet, FBR can occur at any stage of the crop cycle. The results of pre- and post-harvest FBR are significant decreases in crop yield, primarily attributed to reduced bulb size (growth suppression), bulb decay, and a reduction in the shelf life of onions [3]. Depending on the field’s infection severity and the cultivar’s susceptibility degree, yield losses can reach from 39% (in Turkey [4]) to 50% (in Nigeria [5] and India [6]). An additional 12–30% of bulb crop reduction might occur during post-harvest storage [7].

Variability in the disease severity potential is a general characteristic of different species in the *Fusarium* populations. Their influence on host susceptibility at different developmental stages has been extensively documented [8]. Typically, the vulnerability of onions to FBR tends to decrease as seedlings age, but it increases during bulb development and even after harvesting [9]. Currently, research is focused on elucidating the mechanisms underlying divergent responses and the impact of diverse mycotoxins produced by *Fusarium* species in determining disease outcomes. This last aspect is critical due to mycotoxins’ potential health risks to humans and animals [10]. The onion plant’s defense mechanisms toolkit includes genes related to seedling and bulb defense. Their expression varies depending on the aggressiveness of the *Fusarium* isolates [1]. However, these protective measures in susceptible plants are not sufficient to effectively guard against FBR. Conversely, secondary metabolites produced by fungi can have varying effects depending on the stage of the plant host colonization. For instance, fumonisin B1, a toxin, contributes explicitly to virulence during the seedling phase [1].

The pathogenicity of *F. oxysporum* isolates was recently assessed in onion seedlings and mature bulbs [11]. Isolates harboring the Secreted in Xylem (SIX) genes, CRX genes, and C5 gene displayed heightened aggressiveness in both seedlings and mature bulbs. However, even SIX-negative *F. oxysporum* isolates caused a significant reduction in onion seedling emergence, making it challenging to classify *Fusarium* isolates clearly as pathogenic or non-pathogenic. Notably, in mature bulbs, SIX-negative isolates carrying the CRX2 gene exhibited higher virulence than those lacking CRX2 [11].

Despite the widespread adoption of different control measures, the FBR ailment remains a significant obstacle for *Allium* producers worldwide [2]. The traditional chemical approach can be efficient. For example, it was reported that shallot basal rot could be managed effectively by dressing (42–45% yield increase) or dip treatment (44% yield increase) with Prochloraz (Mirage) or the neonicotinoid SeedPlus. Many other compounds were tested against FBR over the years, with varying degrees of success (summarized in [2,12]). In addition, new coping approaches were introduced, including crop rotation, resistant cultivars, and biological control (summarized by [2]). The latest approaches include biocontrol agents (such as members of the fungus genus *Trichoderma* spp. and the *Pseudomonas fluorescens* and *Bacillus* bacteria species), plant extracts, and soil amendments based on natural organic matter (biofumigants or a supplemental source of microbial antagonists to improve the functioning of bio-agents and to stimulate plants’ resistance). Additionally, other less common measures, such as submerging bulbs in hot water, soil solarization, and heating the soil layers on a seedbed by burning plant residues, were tested [2].

Le and colleagues [2], who reviewed the above measures, suggested that timely identification and swift response are crucial to mitigating the negative effects of FBR on onion crops. They conclude that disease control should encompass a comprehensive approach, utilizing cultural, chemical, and biological measures. This may entail using disease-free onion sets for planting, implementing crop rotation to prevent fungus accumulation in the soil, applying fungicides, and employing biocontrol agents. It is essential to explore alternative strategies to expand our FBR response set. This was demonstrated by recent studies focused on breeding for resistance and utilizing RNA interference to silence specific fungal genes [13,14]. Furthermore, implementing proper sanitation measures is essential to prevent fungus spread. These hygiene methods involve removing infected plants and plant remains from the field and disinfecting tools and equipment after each use.

In Israel, FBR was observed in different onion-growing regions across the country [15,16]. Despite recent advancements in scientific research [3,17,18], limited data are available on the disease in Israel. An investigation conducted in 2017–2018 successfully isolated and identified four *Fusarium* species from onion bulbs collected from fields infected with FBR in the Golan Heights region of northeastern Israel [18]. These species include *F. proliferatum*, *F. oxysporum* f. sp. *cepae*, *F. acutatum*, and *F. anthophilium* (the latter two species being less well-known as agents of FBR). However, other pathogenic *Fusarium* species, yet to be identified, may also contribute to FBR. There are significant gaps in our understanding of the nature and distribution of the disease in Israel, as well as limited control methods to combat it. Specifically, there is a lack of structured data on the disease’s historical prevalence, spread rate over time, and current distribution map. Moreover, no onion cultivars resistant to FBR have been identified thus far. Only recently have new fungicides that are effective against the causal pathogens of Israel’s FBR been established, but their application on a commercial field scale has not yet been implemented [3].

Among them, the Prochloraz preparation, combined with irrigation, displayed efficacy against *F. oxysporum* f. sp. *cepae* (B14 isolate) in a semi-field pot experiment that lasted a full season (115 days). The application of Prochloraz prevented the inhibition of sprouting caused by *F. oxysporum* f. sp. *cepae* (B14 strain) in both yellow Orlando (Riverside cv.) and red Noam onion varieties. This protective effect was evident on days 35 and 65 (midseason evaluation) and during harvest (day 115). Notably, even at a low concentration of 0.15%, Prochloraz improved the growth indexes of both onion genotypes under *F. oxysporum* f. sp. cepae stress. However, this compound was less effective against *F. acutatum* [3]. Another preparation, Fludioxonil + Sedaxen (Fl-Se), applied to the seed coating, exhibited potential in protecting both onion cultivars against the two tested *Fusarium* species. These findings highlight the efficacy of Prochloraz and Fl-Se in mitigating the impact of *Fusarium* basal rot disease on onion crops [3].

The current communication report presents a novel quantitative real-time PCR (qPCR)-based approach to monitoring pathogens within the basal plate of the host onion plant under a chemical protective suit. The molecular tracking is compared to a deep and comprehensive analysis of the effects of the experiment’s chemical treatments [3] on plant growth and health indexes. Additionally, the molecular technique provides new sensitive insights into *F. oxysporum* f. sp. *cepae* and *F. acutatum* pathogenesis and interactions under FBR stress and the protective treatments’ impact.

## 2. Materials and Methods

### 2.1. Fungal Species’ Source and Growth Conditions

Two *Fusarium* species were selected for this work, *Fusarium acutatum* (B5 isolate) and *Fusarium oxysporum* f. sp. *cepae* (B14 isolate). These species were previously isolated from diseased onions sampled from commercial fields in the Golan Heights in northeastern Israel [18]. They were identified using colony and microscopic morphology characterization and PCR targeting (followed by sequencing) of the *Fusarium* translation elongation factor-1 alpha gene (TEF1) and the *F. oxysporum* f. sp. *cepae* species-specific putative effector Secreted in Xylem 3 genes (SIX3). Finally, Koch’s postulates approved the identified species as the main causes of onion *Fusarium* basal rot (FBR) using seedling and bulb pathogenicity assays. Their virulence degree and sensitivity to selected fungicides were studied thereafter [3,17].

The growth of the *Fusarium* species in a solid potato dextrose agar (PDA; Difco Laboratories, Detroit, MI, USA) medium was performed by transmitting 6 mm diameter disks from the margins of a 4–6-day-old colony to a new culture plate. The dishes were incubated in the dark at 28 ± 1 °C.

### 2.2. Brief Description of the Semi-Field Trial

This communication report implements new quantitative molecular real-time PCR (qPCR) tracking of the FBR pathogens within the host tissues. It compares a sensitive sensing of infection variations to plant growth and health status under the influence of protective chemical treatments. To this end, we deeply analyzed the large data set collected during a recently reported open-enclosure pot assay [3] and concisely measured the treatments’ significant influence using heat maps. The full-season semi-field trial evaluated new FBR chemical intervention against the disease’s two primary causal agents in northeast Israel, *F. oxysporum* f. sp. cepae and *F. acutatum*. The substances used had previously [17] proved to be effective against the FBR causal agents in the plate inhibition assay: Prochloraz (Pr, demethylation (DMI) inhibitor), Azoxystrobin + Tebuconazole (Az-Te, quinone outside (QoI) inhibitor and DMI-fungicide), and Fludioxonil + Sedaxen (Fl-Se, Phenyl Pyrroles-fungicide and succinate dehydrogenase (SDHI) inhibitor). The trial was conducted at the Avnei Eitan Experimental Farm in the Golan Heights in northeastern Israel. The pots (10-L) were placed in an open area and watered with computerized drip irrigation (1 L/pot daily). The pots were filled with infected ground from two commercial fields in the Golan Heights: from a yellow-onion-growing area, the Orlando variety (Riverside cv., Kibbutz Ortal plot); and from a red-onion-growing area, the Noam variety (Moshav Eliad plot). Perlite number 4 was added (at a ratio of 1:3) to aerate the soil. Five onion seeds were planted in each pot (in the soil of a yellow onion field, Orlando cv.; and in the soil of a red onion field, Noam cv.). Each experimental and control group included 10 pots (biological repetitions), with a total of 260 pots used.

The inoculation procedure consisted, besides naturally infected soil, three steps of complementary inoculation addition: (1) Pre-seeded soil was infected with sterilized pathogen-inoculated wheat grains, prepared as previously described [19,20]. The soil was infected by blending 20 g of the infected sterilized wheat grains with the soil’s top 10 cm layer; (2) two mycelia colony disks (see Section 2.1) from *F. oxysporum* f. sp. *cepae* or *F. acutatum*, or a combination of both pathogens (one disk from each fungus), were added to each plant during the seeding; (3) similarly, two mycelia disks were added to each sprout 21 days after sowing.

The pesticides were applied to the irrigation water (1 L/pot) 16, 35, and 56 days post-sowing. The control was pesticide-free infected soil. The fungicides’ dosages were selected according to the manufacturer’s recommendation. The compounds used were formulated and mixed with water to achieve the desired concentration. The experimental groups that received Prochloraz or Az-Te treatments were divided into two subgroups, each containing five pots. One subgroup received high-dose pesticide treatments (0.3%), while the second received a low dose (0.15%). The Fl-Se preparation was applied via seed coating (0.003 microliters per seed) according to the producer’s specifications.

The semi-field experimental design included complete randomization of the treatments (the location of pots in the farm’s trial area) and statistical analysis of all measures (growth parameters and health indexes) using the JMP program, 15th Edition (SAS Institute Inc., Cary, NC, USA), based on a one-way analysis of variance (ANOVA) with a post hoc comparison based on a *t*-test and a significance level of *p* < 0.05.

### 2.3. Evaluation of Disease Severity under Chemical Control

Growth indices and disease symptoms were evaluated 65 days after sowing and at the season’s end (day 115). The collected growth parameters served as indicators of disease severity and treatment effectiveness, including the number of surviving plants, shoot fresh weight and height, number of leaves, and the weight of onion bulbs. Additionally, the plants’ shoot water content was determined by subtracting their dry weight from their wet biomass weight. The plants’ growth and health results for all experiments were analyzed by calculating the differences in percentages for each index in each treatment in relation to the control—nonprotected infected onion plants. The average measure for all growth and health results was calculated separately. Finally, the degree (rank) of all experiments according to each assay was set and received a matching color in the heat map produced using Microsoft Excel software (Microsoft 365 version).

### 2.4. Real-Time PCR Molecular Evaluation

Molecular tracking of the *Fusarium* spp. infection level within the plants’ basal plates was conducted on both sampling days, 65 and 115 days after sowing. Each plant’s basal plate was thoroughly washed with tap water. Subsequently, it was sliced, and the total weight of each sample was adjusted to 0.7 g. The DNA of the pathogen was isolated and extracted using a previously published protocol [21] with slight modifications [22].

The quantitative real-time PCR method was conducted using an ABI-7900HT device (384-well plates) from Applied Biosystems in Foster City, CA, USA. The technique followed a standard qPCR protocol optimized for detecting fungal DNA instead of mRNA (cDNA), as described in previous studies [22,23,24]. Each qPCR reaction was carried out in a total volume of 5 μL: 0.25 μL of each primer (forward and backward, at a concentration of 10 μM), 2.5 μL of a ready reaction mixture (iTaq™ Universal SYBR Green Supermix solution from Bio-Rad Laboratories Ltd., Hercules, CA, USA), and 2 μL of DNA template. The reaction conditions consisted of an initial denaturation step at 95 °C for 60 s, followed by 40 cycles of denaturation at 95 °C for 15 s, annealing at 59 °C for 30 s, and a final step for creating a melting curve. This last step is 95 °C for 10 s, 65 °C for 31 s, and 60 cycles of 65 °C for 5 s. The Fus-for/rev primers were designed here to amplify a 115 bp oligonucleotide from the *Fusarium* translation elongation factor-1 alpha (TEF1) gene. The COX gene primers target a housekeeping gene coding for cytochrome oxidase in the mitochondria. The primers’ sequences are detailed in Table 1. The ΔCt model was applied to normalize the amount of the *Fusarium* spp. DNA, assuming equal efficacy of amplification for all samples [25,26]. All amplifications were performed with four technical replications.

As described in Section 2.2, the experiment included 5–10 biological replications (pots). On day 65, the plants in each pot were thinned from five plants to one. At that stage, the disease in some treatments was severe and caused seedling death (damping off). Consequently, in some experimental groups (especially in the Azoxystrobin + Tebuconazole-treated pots), only one plant was left for the 65-day sampling. Thus, the midseason result includes only the control treatments, from which 2–7 biological repeats were obtained. The season-ending samples contain 2–10 biological repeats (all surviving plants), and all treatments are included.

### 2.5. Statistical Analysis

Statistical analysis was carried out using the JMP program, 15th Edition. The qPCR results were analyzed using a one-way analysis of variance (ANOVA) with a post hoc comparison based on a *t*-test and a significance level of *p* < 0.05.

## 3. Results

The current communication report aimed to deepen our understanding of the chemical treatments’ impact on *Fusarium* basal rot disease (FBR), presented by us previously [3] (Figure 1). To this end, we introduced a new quantitative real-time PCR analysis of the pathogen within the plants’ basal plates. Pathogenesis and infection severity were compared to the growth and health measurements. Heat maps that emphasize the variations in each index were used to facilitate an overall understanding of the data set. The heat maps enable a comprehensive analysis of all the results and summarize the effect of the treatment on the plants’ growth and health indexes at midseason and on harvest days (65 and 115 days post-sowing).

### 3.1. Evaluation of Disease Severity

The chemical treatments’ impact on *Fusarium* basal rot disease (FBR) was studied by us previously [3]. The numerous data collected and the outcome of these trials were complex to process, and the resulting image may not be fully understood. Thus, a comparative evaluation was carried out here to identify and summarize the various treatments’ main effects. We compared onion growth parameters and physical condition results (in percentages for each index in each treatment) to the control—infected plants without protective intervention (Figure 2 and Figure 3).

In both onion varieties tested, Prochloraz at a low dose (0.15%) significantly excelled in improving the growth and health parameters throughout the season (Figure 2). The compound high dose (0.3%) is probably an overdose that causes a phytotoxic effect and disrupts plant growth (most evident in the yellow Orlando cv. in the *F. acutatum* (B5)- and double-inoculation-treated plants). Interestingly, in the red Noam cv. pots, the Prochloraz high dosage was effective against *F. oxysporum* f. sp. *cepae* (B14) at the sprouting phase (up to 65 days from seeding) but failed to protect the plants at the season’s end (day 115). Contrary to this, Prochloraz’s low dosage was most effective in safeguarding mature onion plants against *F. acutatum,* but not sprouts. In the dual infection treatment, Prochloraz concentrations should be raised towards the season-ending to gain maximum shielding.

The second most efficient compound was Fludioxonil + Sedaxen (Fl-Se, Figure 3). This antifungal mixture performed highly against *F. acutatum* in the mature yellow Orlando cv. plants. Yet it provides solid protection throughout the growth cycle against *F. oxysporum* f. sp. *cepae* in this cultivar and shows promising results against both pathogens in the red Noam cv. Indeed, for this genotype infected by both pathogens, the Fl-Se preparation received the highest and most stable score (Figure 2).

The Azoxystrobin + Tebuconazole (Az-Te) pesticide was the less beneficial treatment in terms of reducing FBR (Figure 3). It appears that the higher dosage of this mixture caused phytotoxicity. Still, the Az + Te low concentration positively influenced the yellow Orlando cv. mature plants against *F. acutatum* and the young plants against *F. oxysporum* f. sp. *cepae*.

Analyzing only the control (highlighted in black at the bottom of Figure 1B and Figure 2A) provides a clear understanding of the pathogens’ aggressiveness and co-interactions. *F. acutatum* is a major threat to the yellow Orlando cv., while *F. oxysporum* f. sp. *cepae* is the primary threat to the red Noam cv. Double inoculating the plants with both pathogens represses the disease in the first cultivar but maintains the high aggression of *F. oxysporum* f. sp. *cepae* towards the second cultivar (Noam cv.).

Most intriguingly, it can be interpreted that, despite the antagonism that suppresses the disease in the yellow Orlando cv., the chemical treatments were less effective in the dual-infection pots; thus, the generally stronger FBR outburst measured in the chemically treated plants (Figure 2). Still, the total summary of the results of the chemical treatments (Figure 3) reveals that the disease severity in the red Noam cv. was higher (indicated by higher numbers in most treatments). This finding is also supported by the plants’ visible symptoms during the harvest (Figure 1 and Figure 4).

### 3.2. Real-Time PCR Molecular Evaluation

Sensitive molecular detection of the *Fusarium* pathogens inside the plants’ basal plates uncovered the full impact of the chemical treatments (Figure 5) and matched the symptom analysis (Figure 2 and Figure 3). When the nonprotected controls’ pathogen DNA was high, all chemical interventions substantially reduced the infection levels, ranging from 34% to 97%. On two occasions, the nonprotected controls’ pathogen DNA was markedly low: in the yellow Orlando cv. dual-infection treatment (B5 + B14) and in the red Noam cv. *F. Acutatum* (B5) treatment. This finding fully overlaps with the growth and health indexes described in the heat map in Figure 2 (green cells). The reason for these results is that pathogenic antagonism exists in the Orlando cv. and the low infection ability of *F. acutatum* in the Noam cv., as will be explained in the Discussion section. Curiously, these tendencies were measured during the harvest, while during the midseason sampling, the infection degree was relatively smaller, as expected, but unexpectedly similar in all nonprotected controls (Figure 6).

## 4. Discussion

*Fusarium* basal rot (FBR) is a major concern of onion growers in Israel and worldwide. Despite significant progress in understanding this disease and developing diverse ways of restricting its harmful impacts, there is still a lack of control protocols. Previous studies have reported variances in the aggressiveness levels of *Fusarium* species that are populating diseased onion plants [2,4]. These include distinct species with a large diversity in terms of specific host ranges and non-pathogenic forms [29]. Differences in response to resistant selections could be attributed to variations among *Fusarium* isolates [30]. The formae specialis *cepae* is one of the host-specific groups within *F. oxysporum,* which is the most acknowledged cause of FBR. Global studies suggest that changes in the species composition of the *Fusarium* species involved in FBR require tailored pest control solutions, as each species may respond differently to fungicidal treatments.

Currently, one of the main goals of the international scientific effort is to develop an eco-friendly biological interphase against FBR. This strategy is harmless to the soil-beneficial microflora and hazard-free to humans and farm animals [12,31]. Moreover, environmentally safe approaches overcome one of the major concerns in chemical pesticide use—the development of fungicidal resistance that would disarm the ability of those compounds to protect plants [32,33,34,35]. Bio-friendly methods rely on bacteria and fungi originating from the soil [36], seeds [37], and other sources such as marine sponges [38]. While such methods gradually enter the commercial market, one weak point concerns the users. The biocontrol species’ functioning is subjected to uncontrollable environmental conditions, so their disease management capacity is often incomplete or unstable.

The traditional chemical approach is still our most powerful and consistent strategy for protecting field crops against phytopathogens, particularly in severe cases. We previously tested diverse chemo-pesticides in the lab using plate screening. We applied selected compounds in a full-season, open-enclosure pot trial to protect two leading commercial onion hybrids [3]. Here, we report new quantitative real-time PCR (qPCR) monitoring to study the pathogens’ infection levels under the stress of these chemicals. Such a sensitive tool allows us to identify potential preventive treatments and deepen our understanding of their mode of operation. The qPCR-based analysis was compared to a new heat map analysis of the plants’ growth and health results. The findings disclose interesting conclusions that were difficult to identify in our former work [3].

The most prominent findings drawn from the new analysis are variations in terms of the effectiveness of each treatment regarding the pre-maturating development stage and the bulbs’ ripening stage. For instance, in the yellow Orlando cv. (Figure 2A), the impact of the Fludioxonil + Sedaxen (Fl-Se) treatment against *F. acutatum* (alone or in a mixture with *F. oxysporum* f. sp. *cepae*) occurred during the plants’ final growth stage. In contrast, Azoxystrobin + Tebuconazole (Az-Te at low dosage) worked better during the plants’ sprout stage against *F. oxysporum* f. sp. *cepae*. This discovery has significant implications because it suggests combining both fungicides (simultaneously or in a sequence) to provide a comprehensive response to the disease throughout the plant lifecycle. Furthermore, the Azoxystrobin component acts as a QoI fungicide (quinone outside inhibitor), an action mechanism that does not exist in the other pesticides tested here. Thus, it should be tested solely against the FBR pathogen and considered an additional treatment to support the other, more successful compounds (Prochloraz and Fl-Se), which have different action mechanisms.

Another example is the Prochloraz treatment in the red Noam onion genotype (Figure 2B). In *F. acutatum* and *F. oxysporum* f. sp. *cepae* mix infection, a low dose of this compound was most effective in young plants. In contrast, the high concentration was more effective in protecting mature plants. Thus, increasing the Prochloraz dosages throughout the season should be considered. This higher dosage is required because of the synergistic relationships between those two *Fusarium* species, causing more severe FBR in the Noam cv., as discussed below.

The qPCR pathogen tracking within the plants’ basal plates (Figure 5) showed that all chemical treatments tested here led to drastic repression of infection levels. This is true even for the treatments that appeared to be ineffective (such as Az-Te, especially in the high concentration applied) in terms of growth promotion or disease symptom elimination. From these results, it can be concluded that the pesticide was effective, but most probably phytotoxic to the plants, and thus lower dosages must be considered. To support this observation, in the red Noam cv. at the season’s end (Figure 2B), in the plants infected by *F. oxysporum* f. sp. *cepae*, this treatment received the lowest marks in growth parameters but remarkably high scores in the health index.

Effective control methods can be adopted from other countries, but they must be evaluated and validated against the local population of *Fusarium* in Israel. For example, in Finland, despite chemical treatments, the incidence of *Fusarium* spp.-infected bulbs can be as high as 20% [39]. This result could be attributed to significant variations in *Fusarium* populations and environmental factors. Therefore, it cannot be assumed that the same control method will be equally effective in all locations. Field evaluation of disease-preventing treatments that test the short-term response of the *Fusarium* species involved in FBR should consider the long-range influence. These treatments can impact the soil and plant microflora communities (including soil recolonization) [40], altering their composition and, consequently, affecting their control efficiency.

The qPCR-based method presented here allows us to closely monitor the pathogenesis under chemical-preventing treatment. It enables us to detect the impact of control interventions, even at the early latent phase of the disease, where no symptoms are apparent. Such benefit was demonstrated by us previously in maize [23,41] and cotton [42]. Thus, the qPCR method can save time and effort. A scientific program to develop disease control cannot depend solely on a full growth period under field conditions. The growing season can last several months and requires significant effort while exposing crops to environmental changes that can lead to inconsistent results. The ability to sensitively identify high-potential protective treatment in sprouts (preferably under controlled conditions) can have a game-changing influence on crop protection research.

Another significant benefit of the new qPCR-based approach for such studies is its ability to track the pathogens’ infection variations and to study their relationships during the onion growth season. Such essential information is revealed here. The untreated controls demonstrated the high infection ability of *F. oxysporum* f. sp. *cepae* compared to *F. acutatum* in both cultivars. Additionally, antagonism between these species led to lower DNA levels for both species in the yellow Orlando cv. Competition for resources such as food and growing space can lead to these antagonistic interactions. Instances of such phytopathogen cross-talk are documented in the literature. For example, in cotton plants, there are negative interactions between *Magnaporthiopsis maydis* and *Macrophomina phaseolina*, the causative agent of charcoal rot disease [43,44]. Additionally, *M. maydis* is also associated with *Fusarium oxysporum*. However, the outcomes of these interactions are not always inhibitory, as *M. maydis*, along with *Fusarium verticillioides* and *M. phaseolina*, can cause severe post-flowering stalk rot diseases in maize [45,46,47]. Another example involves the cross-talk in the pathobiome of pea roots [48,49,50], where the co-occurrence of plant pathogens affects their development and disease severity. Still, the co-existence of pathogens within a disease complex can favor plant damage and pathogen reproduction [44,49], depending on the sequence of inoculation events, environmental conditions, and host susceptibility.

## 5. Conclusions

*Fusarium* basal rot disease (FBR) poses a destructive threat to onion crops worldwide. It has emerged as a significant disease in Israel and urgently requires new coping strategies. In a recent full-season pot experiment, we investigated the chemical protection of two commercial onion cultivars against FBR. Here, we employed new real-time PCR (qPCR) molecular tracking to monitor the pathogens in the host plant’s basal plate and conducted a comprehensive analysis of the experimental treatments’ effects on plant growth and health. The molecular-based analysis supported the observed susceptibility variations and confirmed the treatments’ impact. Implementing Prochloraz or Fludioxonil + Sedaxen (Fl-Se) reduced root infection by 80–90% in most cases. Moreover, the results imply that less beneficial compounds in terms of growth promotion, such as the Azoxystrobin + Tebuconazole (Az-Te), are due to high toxicity rather than lack of efficiency. Prochloraz at low concentrations protects against *Fusarium oxysporum* f. sp. *cepae* throughout the crop cycle in the yellow Orlando and the red Noam cultivars, yet it failed to protect Noam cv. plants from *F. acutatum* at the sprouting phase. The Fl-Se mixture had a similar FBR safeguarding impact in both onion genotypes, yet, unlike Prochloraz, it could shield only mature Orlando cv. plants from *F. acutatum*. On the other hand, the Az-Te (low dosage) preparation exhibited superior effectiveness against *F. oxysporum* f. sp. *cepae* during the plants’ sprout stage. Thus, a combination of Az-Te and Fl-Se may provide a wide-ranging and more robust shield during the Orlando cv. season. The unprotected controls uncovered the high infection ability of *F. oxysporum* f. sp. *cepae* compared to *F. acutatum* in both onion genotypes and antagonistic interactions between these species that led to their DNA levels decreasing in the yellow Orlando cv. Our findings demonstrate that molecular tracking is a valuable tool for evaluating the efficiency of protective protocols and identifying ways to enhance their impact. Future studies should test the qPCR-based approach in a seedlings assay in controlled conditions and on a field scale for an entire season to unravel its full potential. Also, it will be important to develop a qPCR analysis that can sensitively distinguish between the *Fusarium* species within the FBR *Fusarium* complex inhabiting the host plant.

## Figures and Tables

**Figure 1 jof-09-00809-f001:**
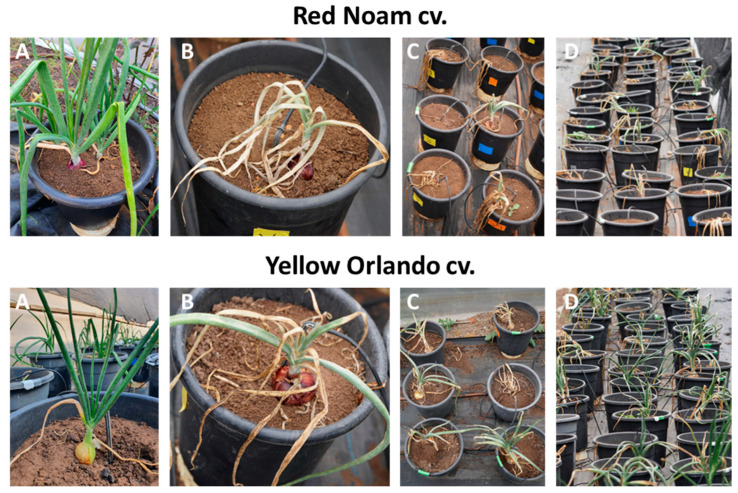
Experiment photos during the harvest (115 days after soil surface emergence). The data were collected in a previous study [3]. Two onion varieties, red Noam cv. (**upper panel**) and yellow Orlando cv. (Riverside cv., **lower panel**), were tested in a pot trial under field conditions at the Avnei Eitan Experimental Farm in the Golan Heights (northeastern Israel). (**A**). A relatively healthy plant. (**B**). A diseased plant. (**C**,**D**). Overview of treatments revealing the more severe disease in the red Noam genotype. The pots were separately infected with two pathogens, *F. oxysporum* f. sp. *cepae* (B14 isolate) and *F. acutatum* (B5 isolate), and were tested against a combination of both pathogens and an unprotected control. Three pesticides, namely Prochloraz (Pr), Azoxystrobin + Tebuconazole (Az-Te), and Fludioxonil + Sedaxen (Fl-Se), were inspected as protective treatments. Each fungicide and control group was replicated ten times. The Prochloraz and Az-Te treatments were further divided into two subgroups of five pots each, receiving a high-dose (0.3%) or a low-dose (0.15%) pesticide treatment applied 16, 35, and 56 days after sowing. The Fl-Se preparation was applied through seed dressing at a rate of 0.003 microliters per seed.

**Figure 2 jof-09-00809-f002:**
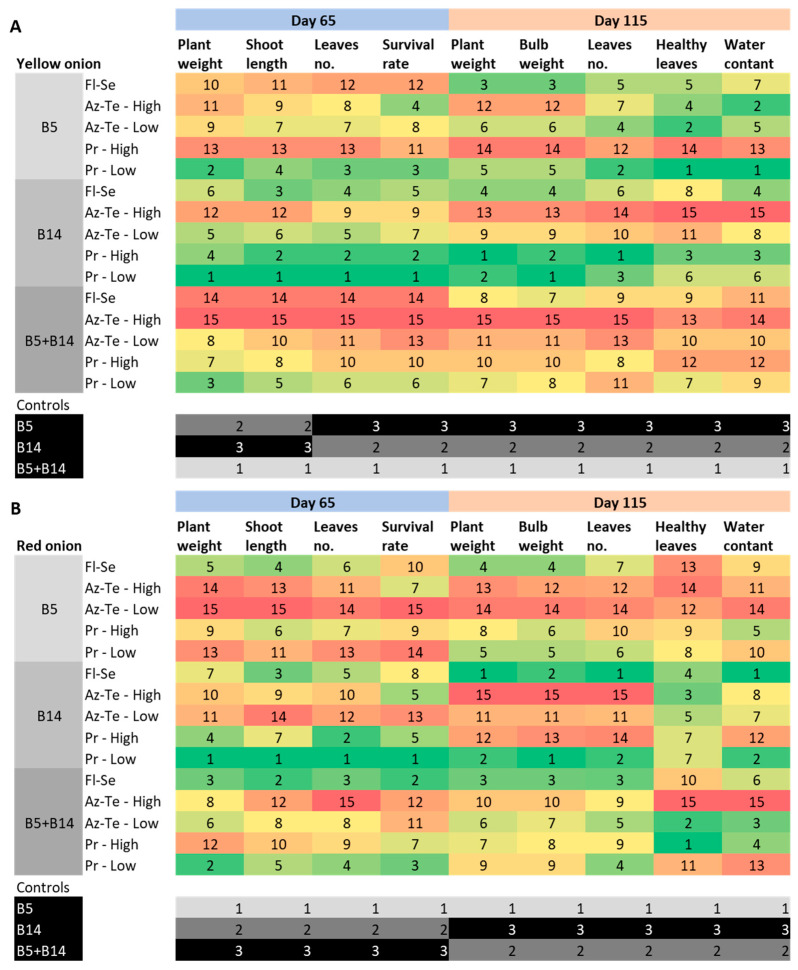
Comparative analysis of the results of the chemical treatments against *Fusarium* basal rot in onions. The data were collected in a previous study [3]. Two onion varieties, yellow Orlando cv. (**A**) and red Noam cv. (**B**), were tested in the semi-field pot trial described in Figure 1. The plants’ growth and health results for all experiments were analyzed by calculating the percentage differences for each index between each treatment and the control, which consisted of infected, nonprotected onion plants. Each index’s rank (degree) was set and received a matching color in the heat map. The best treatments in terms of growth and health score are placed at the top of the ranking (numbers 1–7) and are signified by a range from green to yellow, while plants affected by the disease (in growth or health) received a low order (numbers 8–15) and are represented by warm colors orange–red). An impact comparison between the controls is presented in the lower sections of the figure (**A**,**B** separately). In this section, the rank is between 1–3 since only three check treatments exist. A grayscale color map describes the differences.

**Figure 3 jof-09-00809-f003:**
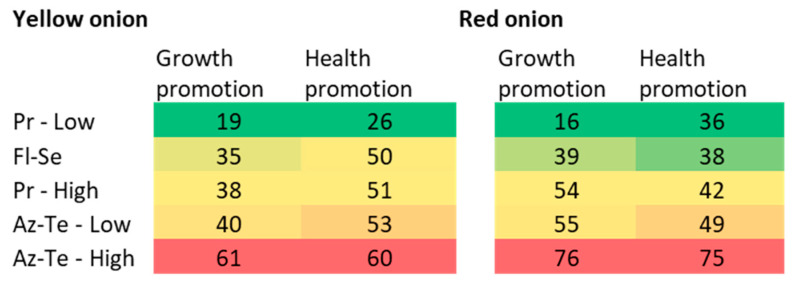
Total summary of the results of the chemical treatments against *Fusarium* basal rot in onions. The average of all measured scores (displayed in Figure 2) is presented. The best treatments regarding growth and health score are placed at the top of the ranking (numbers 16–40) and are signified by color map ranges from green to yellow. Plants affected by the disease (in growth or health) are at the bottom (numbers 41–76) and are represented by warmer colors (orange–red).

**Figure 4 jof-09-00809-f004:**
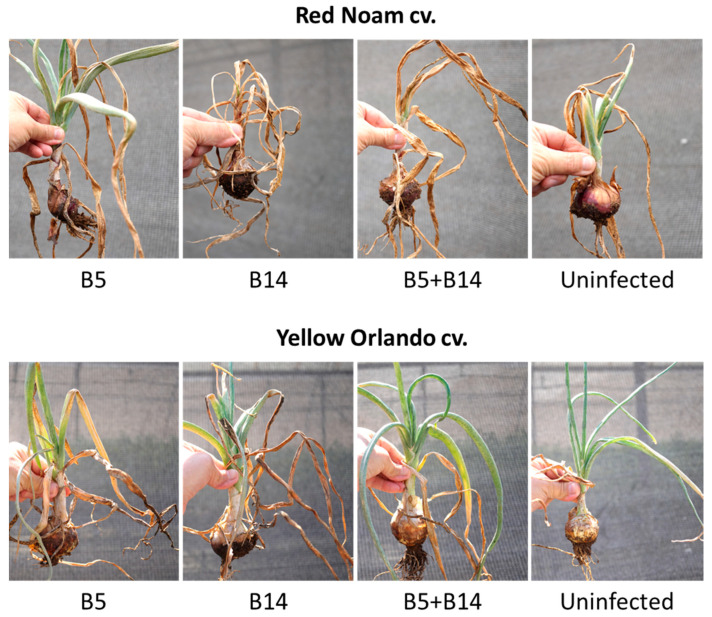
The visible symptoms of the infected untreated plants from the semi-field experiment during the harvest (day 115). Representative photos of the control (nonprotected onion plants) demonstrate the plants’ growth and health variations under the stress of the *Fusarium* basal plate pathogens. The full experiment is described in Figure 1. The pots were infected (separately or combined) with two pathogens, *F. oxysporum* f. sp. *cepae* (B14 isolate) and *F. acutatum* (B5 isolate). Uninfected plants were grown on naturally infested soil without complementary soil inoculation.

**Figure 5 jof-09-00809-f005:**
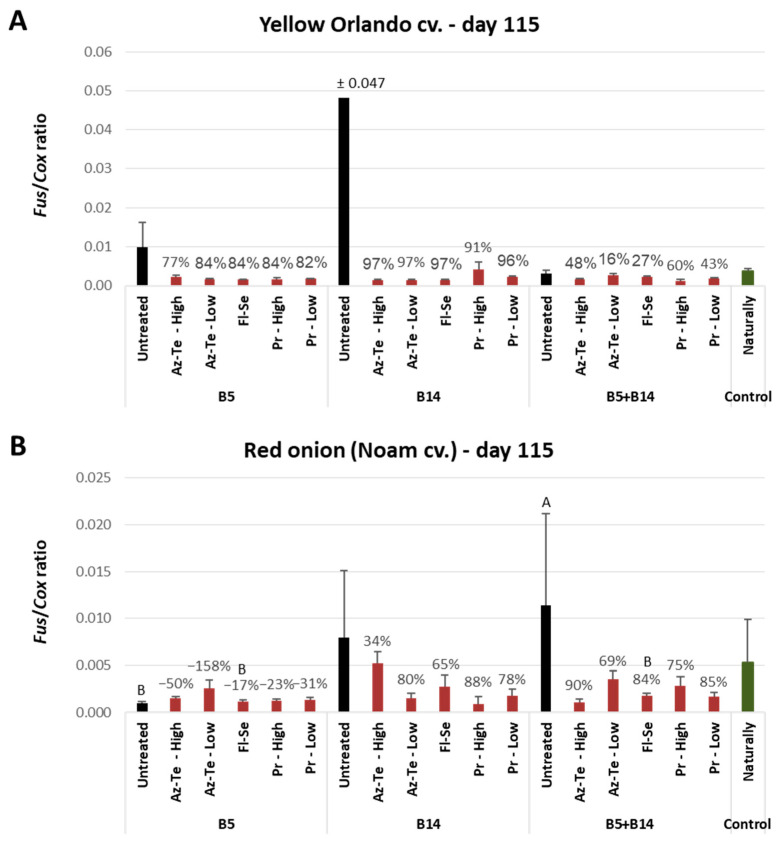
Molecular real-time PCR-based analysis of the results of the chemical treatments against *Fusarium* basal rot in onions. The experiment is described in Figure 1. The evaluation was made in the open-enclosure trial during the harvest (day 115) for the yellow Orlando cv. (**A**) and the red Noam cv. (**B**). All pots included naturally-infested soil from commercial fields. In the right column (the control, highlighted in green) are plants that grew in such soil. The other plants were additionally inoculated by the pathogens *F. oxysporum* f. sp. *cepae* (B14 isolate) and *F. acutatum* (B5 isolate), separately or combined (unprotected columns highlighted in black). The relative amount of *Fusarium* spp. DNA (*Fus*) normalized to the cytochrome C oxidase DNA (*Cox*) is presented on the Y axis. Each value is a mean of 2–7 repetitions (plants per treatment). Error bars indicate the standard error. If a significant difference is identified (ANOVA test, *p* < 0.05), different letters (A,B) above the chart’s bars are presented. Otherwise, no statistical significance was found.

**Figure 6 jof-09-00809-f006:**
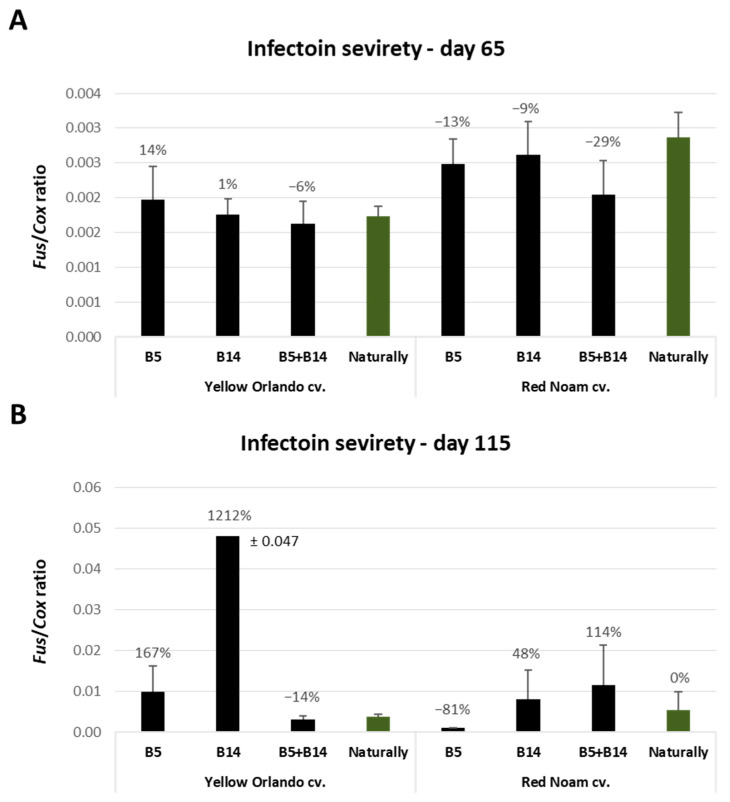
Molecular analysis of the *Fusarium*-infected unprotected plants (the controls). The experiment is described in Figure 1. The molecular examination and axis explanation appears in Figure 5. In the right column (the control, highlighted in green) are plants that grew in naturally-infested soil from commercial fields. The other plants were additionally inoculated by the pathogens *F. oxysporum* f. sp. *cepae* (B14 isolate) and *F. acutatum* (B5 isolate), separately or combined. Each value is a mean of 2–7 ((**A**), midseason) or 2–10 ((**B**), season-ending) repetitions (plants per treatment). Error bars indicate the standard error.

**Table 1 jof-09-00809-t001:** Primers for *Magnaporthiopsis maydis* detection.

Pairs	Primer	Sequence ^1^	Uses	Amplification	References
Pair 1	Fus-for Fus-rev	5′-CGACCACTGTGAGTACTACCATC-3′ 5′-ACCGGTCTGTCAAGCTATGT-3′	Target gene	*Fusarium* spp.-specific fragment, qPCR cycling—27 or above	This work
Pair 3	COX-F COX-R	5′-GTATGCCACGTCGCATTCCAGA-3′ 5′-CAACTACGGATATATAAGRRCCRRAACTG-3′	Control	Cytochrome C oxidase (COX) gene product, qPCR cycling—27 or below	[27,28]

^1^ The R symbol represents Guanine or Adenine (purine). The synthesized primer contained a mixture of primers with both nucleotides.

## Data Availability

The data sets generated and/or analyzed during the current study are available from the corresponding author upon reasonable request.

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
