# Peer review of "Molecular Real-Time PCR Monitoring of Onion Fusarium Basal Rot Chemical Control"

_jof, 2023, doi:10.3390/jof9080809_

Round 1

Reviewer 1 Report

-> Were used keywords that are already contained in the title (eg.: basal rot; chemical control, Fusarium; onion,  Real time PCR). 

-> In 2.2. "Brief description of the semi-field trial": it is not clear which statistical design was used for the pot test. It probably has a more appropriate statistical procedure than excel to analyze the data, however the design is not clear. 

-> Line 257 Figure 1: The best treatments in terms of growth and health score are placed at the top of the ranking (numbers 1-7) and are signified in green to yellow, while plants affected by the disease (in growth or health) received a low order (numbers 9-15) and are represented by warm colors (orange-red).” However, in the controls the numbers “3” are in orange, which makes the figure not self-explanatory.

-> Line 274: "In both onion varieties tested, Prochloraz dominantly excelled in improving the 274 growth and health parameters throughout the season. This is true for the high- 275 dose (0.3%) or low-dose (0.15%) ". However in figure 2 we see that "Fl-Se" was better than "Pr-Hight". This must to be better explained.

-> Figure 3 should be self-explanatory and not refer to another figure. Furthermore, throughout the paper, we understood that control plants were plants that did not undergo chemical treatment to control the disease. But in figure 3 the best plant is "Control" and it makes us understand that all the photos are of “control” plants and the photo that is called “control” must have been an uninfected plant. If this is not true, it is necessary to inform, for both cultivars, which were the chemical controls used in photos B5, B14, B5+14.

The analysis of figures 4 and 5 left the question of how efficient the real time PCR was in quantifying the pathogen since the control was not efficient in two cases. Or it remains in doubt whether the inoculation of the pathogens was really effective. Could this same "error" have occurred in some of the chemical treatments evaluated?

Author Response

Responses to Reviewer 1's comments

We thank the reviewer for investing substantial efforts, which undoubtedly contribute to this manuscript. All your raised points were addressed carefully and thoroughly, as detailed below. The remarks and suggestions improved this paper's scientific soundness and accuracy. Your contribution is greatly appreciated.

  1. Were used keywords that are already contained in the title (e.g., basal rot, chemical control, Fusarium, onion,  Real-time PCR).

 Reply: All keywords that are already contained in the title were replaced.

  1. In 2.2. "Brief description of the semi-field trial": it is not clear which statistical design was used for the pot test. It probably has a more appropriate statistical procedure than Excel to analyze the data; however, the design is not clear. 

Reply: The reviewer is correct; the semi-field trial statistical design should be explained better. We added the following information to the text (section 2.2, lines 184-188): "The semi-field experimental design included complete randomization of the treatments (pots locations in the farm's trial area) and statistical analysis of all measures (growth parameters and health indexes) using the JMP program, 15th Edition (SAS Institute Inc., Cary, NC, USA) based on one-way analysis of variance (ANOVA) with a post-hoc comparison based on a t-test and a significance level of p < 0.05."

  1. Line 257 Figure 1: The best treatments in terms of growth and health score are placed at the top of the ranking (numbers 1-7) and are signified in green to yellow, while plants affected by the disease (in growth or health) received a low order (numbers 9-15) and are represented by warm colors (orange-red)." However, in the controls, the numbers "3" are in orange, which makes the figure not self-explanatory.

Reply: Indeed, this should be explained better. In the above part (the main figure body), each treatment impact was compared to the control and received rank accordingly. This was already explained in the figure (now Figure 2) legend (lines 277-281): "The plants' growth and health results in all experiments were analyzed by calculating the percentage differences of each index between each treatment and the control, which consisted of infected, nonprotected onion plants. Each index's rank (degree) was set and received a matching color in the heat map."

For that reason, we couldn't include the control groups in this analysis. Therefore we present an impact comparison between the controls in the lower section of the figure (A and B separately). In this section, the rank is between 1-3 since only three check treatments exist. Indeed the colors were confusing; thus, we chose grayscale colors to describe the differences.

  • The figure (now Figure 2) was replaced with an improved figure.
  • The following explanation was added to the Figure 2 legend (lines 284-286): "An impact comparison between the controls is presented in the lower sections of the figure (A and B separately). In this section, the rank is between 1-3 since only three check treatments exist. A grayscale color map describes the differences."
  1. Line 274: "In both onion varieties tested, Prochloraz dominantly excelled in improving the growth and health parameters throughout the season. This is true for the high-dose (0.3%) or low-dose (0.15%) ". However, in Figure 2 we see that "Fl-Se" was better than "Pr-Hight". This must be better explained.

Reply: This is an important and correct remark. Thank you. The paragraph was edited to refer only to the Prochloraz low dose (0.15%) as the best treatment. Also, the Fl-Se was stated as the second-best treatment. The Prochloraz high concentration (0.3%) is probably an overdose that causes a phytotoxic effect that disrupts plant growth.

The above explanation was embedded in the text:

  • Lines 287-291: "In both onion varieties tested, Prochloraz at a low dose (0.15%) dominantly excelled in improving the growth and health parameters throughout the season (Figure 2). The compound high-dose (0.3%) is probably an overdose that causes a phytotoxic effect and disrupts plant growth (most evident in the yellow Orlando cv. in the F. acutatum (B5) and the double inoculation-treated plants)."
  • Line 296: "The second most efficient compound was Fludioxonil + Sedaxen (Fl-Se, Figure 3)."
  1. Figure 3 should be self-explanatory and not refer to another figure. Furthermore, throughout the paper, we understood that control plants were plants that did not undergo chemical treatment to control the disease. But in Figure 3 the best plant is "Control" and it makes us understand that all the photos are of "control" plants and the photo that is called "control" must have been an uninfected plant. If this is not true, it is necessary to inform, for both cultivars, which were the chemical controls used in photos B5, B14, B5+14.

Reply: Correct; the figure (now Figure 4) legend needs to be clearer and more precise. Indeed in Figure 4, all the photos are of plants that didn't receive any protective chemical treatment. The picture called "control" is an uninfected plant. Thus, the following changes were made:

  • We replace Figure 4; in the new figure, we replace the term "Control" with the word "Uninfected."
  • Figure 4 legend was rewritten and now reads: "Figure 4. The semi-field experiment infected untreated plants' visible symptoms at the harvest (day 115). Representative photos of the control (nonprotected onion plants) demonstrate the plants' growth and health variations under the Fusarium basal plate pathogens' stress. The full experiment is described in Figure 1. The pots were infected (separately or combined) with two pathogens, F. oxysporum f. sp. cepae (B14 isolate) and F. acutatum (B5 isolate). Uninfected plants were grown on naturally infested soil without the complimentary soil inoculation."
  1. The analysis of Figures 4 and 5 left the question of how efficient the Real-Time PCR was in quantifying the pathogen since the control was not efficient in two cases. Or it remains in doubt whether the inoculation of the pathogens was really effective. Could this same "error" have occurred in some of the chemical treatments evaluated?

Reply: Thank you for this important remark. Indeed, as already mentioned in the text (lines 334-337): " On two occasions, the nonprotected controls' pathogen DNA was markedly low: in the yellow Orlando cv. dual infection treatment (B5+B14) and in the red Noam cv. F. acutatum (B5) treatment. This finding fully overlaps the growth and health indexes described in Figure 2 heatmap (green cells)."

The reason for these results is the antagonism between F. oxysporum f. sp. cepae and F. acutatum that exist in the Orlando cv. and the low infection ability of F. acutatum in the Noam cv.

The explanation for these results was already elaborated in the:

  • Abstract (lines 24-26): "The molecular-based analysis discloses the high infection ability of F. oxysporum f. sp. cepae compared to F. acutatum in both cultivars. It also indicates an antagonism between those species in the Orlando cv."
  • Conclusion (lines 440-443): "The untreated controls demonstrated the high infection ability of F. oxysporum f. sp. cepae compared to F. acutatum in both cultivars. Additionally, antagonism between these species led to lower DNA levels of both species in the yellow Orlando cv."

Nonetheless, we :

  • Added a short explanation in the result section as well, to ensure to the reader that the results are accurate and true (lines 337-339): "The reason for these results is pathogens antagonism exists in the Orlando cv. and the low infection ability of F. acutatum in the Noam cv., as will be explained in detail in the Discussion."
  • We added the clarification and new information from the literature in the Discussion section to explain the results (lines 448-455): "Another significant benefit of the new qPCR-based approach for such studies is its ability to track the pathogens' infection variations and to study their relationships during the onion growth season. Such essential information is revealed here. The untreated controls demonstrated the high infection ability of F. oxysporum f. sp. cepae compared to F. acutatum in both cultivars. Additionally, antagonism between these species led to lower DNA levels of both species in the yellow Orlando cv. Competition for resources such as food and growing space can lead to antagonistic interactions. Instances of such phytopathogens cross-talk are documented in the literature. For example, in cotton plants, there are negative interactions between Magnaporthiopsis maydis and Macrophomina phaseolina, the causative agent of charcoal rot disease [44,45]. Additionally, M. maydis is also associated with Fusarium oxysporum. However, the outcomes of these interactions are not always inhibitory, as M. maydis, along with Fusarium verticillioides and M. phaseolina, can cause severe post-flowering stalk rot diseases in maize [46-48]. Another example involves the cross-talk in the pea roots pathobiome [49-51], where these plant pathogens' co-occurrence affects their development and disease severity. Still, the co-existence of pathogens within a disease complex can favor plant damage and pathogens' reproduction [45,50], depending on the sequence of inoculation events, environmental conditions, and host susceptibility."

Reviewer 2 Report

The paper by Dimant and Degani describes the ability of pesticides to fight against the fusarium basal rot disease (FBR). The authors use five chemical fungicide treatments on two onion varieties infected with B5 and B14 isolates. 

The paper is well written and the work provides illustrations of onion plants to visualize the impact of B5, B14 and B5+B14 infections. These illustrations give strength to the data but the figure 3 would need additional illustrations about the comparative beneficial effect of the various chemical treatments. An illustration of the semi-field trials would be also beneficial.

This paper is mostly descriptive data, a lot of work on biometric measurements. Fungicide treatments are carried out in semi-field conditions. High dose (0.3%) of pesticide is twice the low dose (0.15%). The authors should explain why these two specific concentrations are used? Are the pesticides water soluble? Are they used formulated or pure? Please give these informations in Materials and Methods section.

The abstract is too long. Please select the most prominent data. Figure 5: beware of typing erros: “infection severity“

Author Response

Responses to Reviewer 2's comments

We would like to express our sincere appreciation to the reviewer for the essential and helpful advice. The time and effort invested are greatly appreciated and certainly contributed to the manuscript and improved it. Thank you.

  1. The paper by Dimant and Degani describes the ability of pesticides to fight against the Fusarium basal rot disease (FBR). The authors use five chemical fungicide treatments on two onion varieties infected with B5 and B14 isolates.

Reply: Thank you for the evaluation of our manuscript. All your remarks and suggestions were addressed carefully and thoroughly, as detailed below.

  1. The paper is well written and the work provides illustrations of onion plants to visualize the impact of B5, B14 and B5+B14 infections. These illustrations give strength to the data but the figure 3 would need additional illustrations about the comparative beneficial effect of the various chemical treatments. An illustration of the semi-field trials would be also beneficial.

Reply: Thank you for these remarks. We only took photos of the control plants individually, so, unfortunately, we can not provide illustrations about the comparative beneficial effect of the various chemical treatments. However, we do have pictures of the semi-field trials.

A new figure, Figure 1, was added. The figure shows the experiment photos at the harvest (day 115 from soil surface germination).

  1. This paper is mostly descriptive data, a lot of work on biometric measurements. Fungicide treatments are carried out in semi-field conditions. High dose (0.3%) of pesticide is twice the low dose (0.15%). The authors should explain why these two specific concentrations are used? Are the pesticides water soluble? Are they used formulated or pure? Please give these informations in Materials and Methods section.

Reply: Indeed, these are important aspects that need to be addressed.

The following explanation was added regarding the questions you raised (lines 177-179): " According to the manufacturer's recommendation, the fungicides' dosages were selected. The compounds were used formulated and mixed with water to achieve the desired concentration."

  1. The Abstract is too long. Please select the most prominent data.

Reply: The reviewer is correct. The Abstract was re-edited, focused, and shortened from 373 to 315 words. We believe all remaining information is crucial to summarize this work's background, aims, methodology, results, and conclusion.

Figure 5: beware of typing erros: "infection severity"

Reply: The figure (now Figure 6) was corrected and replaced. Thank you.

Round 2

Reviewer 1 Report

The paper is clearer and more informative. It's much better than the previous version.

I recommend removing "as will be explained in detail in the Discussion " on line 338.